# Slow Collecting: Sustainability and the Need for a Paradigm Shift by Iberian Collectors

**Adelaide Duarte** [1,*,†] and **Marta Pérez-Ibáñez** [2,*,†]

1   Art History Institute, Universidade NOVA de Lisboa—School of Social Sciences and Humani-ties/IN2PAST—Associate Laboratory for Research and Innovation in Heritage, Arts, Sustainability and Terri-tory, Campus de Campolide 1099-032 Lisboa, Portugal
2   Facultad de Ciencias de la Comunicación, Universidad Francisco de Vitoria, 28223 Madrid, Spain
*   Correspondence: duarte.adelaide@gmail.com (A.D.); martaperezib@gmail.com (M.P.-I.)
†   Both authors are members of the association TIAMSA.

**Abstract:** Collectors are major actors in the global art market as they often spend large sums of money fostering the business. Concerning sustainable collecting practices—i.e., the balance between what is the best for people and for the environment—collectors' actions seem contradictory. Firstly, onto-logically, to collect is to accumulate artworks; secondly, art—the object of the collectors' desire—and the global art world are not closely aligned with the climate crisis. The art ecosystem encourages trips to participate in art events worldwide, increasing the carbon footprint impact, and rarely uses recycled materials, causing waste. The economic model of the art market lacks sustainability, raising the question: how can we promote a sustainable collecting attitude? In this exploratory study, we will observe art market players, especially the Iberian Peninsula collectors' actions, in terms of their contribution to reducing the environmental impact of purchases. Based on data, reports, interviews, and published sources, we will investigate collectors' awareness of the subject and evaluate their adopted actions. As, to date, no analysis has been carried out on the trends of Iberian collecting in the field of climate sustainability, we have focused our study on finding data from the primary source par excellence: the collectors themselves. The aim is to fuel the need for a paradigm shift, concluding on a slow collecting attitude.

**Keywords:** slow collecting; art markets; Iberian collectors; contemporary art; sustainable collecting

## 1. Introduction

Sustainability is at the forefront of debate at present, as the rapid climate changes are making the issue urgent. Concerning the global art market's performance in relation to sustainability, its specific way of functioning—depending on a strong calendar of events, such as art fairs or exhibitions worldwide—makes the topic particularly challenging. On the other hand, the ability of the field to search for alternative measures, and the resistance to external shocks, has been a way of overcoming the complex issues in the sector [1]. In addition to the recent health pandemic crisis and the ongoing consequences of the Russian invasion of Ukraine, the art trade has found solutions as exploring hybrid models for buying and selling art, showing a prompt adaptation to the digital trade, and the online platforms developed by galleries and art fairs became more usual. In terms of the estimated global figures, the global art market reached an estimated USD 67.8 billion in 2022, according to McAndrew's report, with global sales increasing by 3% compared to the previous year [2]. Regarding sustainability, however, and in consonance with the Art of Zero report launched a year before, in 2021, an estimate of the global carbon footprint for the visual arts sector in the order of 70 million tons of CO2e per year was presented, considering that the major percentage of the sector's figures is due to visitor travel (circa 74% of those figures) [3]. These data mean that the visual art sector needs a stronger commitment to the transition to net zero carbon, as it is still a very polluting industry.

Nevertheless, the infrastructures of the art market are showing concern with the urge of climate changes. Museums have been discussing sustainability for a long time, alongside the potential impact of the collections' growth and how to take care of them, from the eco-museums in the seventies to the museum's performance in the climate crisis [4–6]. Art fairs are presently encouraging the greening up of its carbon-heavy footprint. London Frieze Art Fair and Art Basel, two leading art fairs for the primary art market, are setting up individual measures in response to climate changes, such as the use of reused energy, biofuel, and the hybrid power model, LED lighting for gallery booths, catering with green credentials, preferably shipping by train or boat [7]. Non-profit associations are also bringing the subject to discussion. In Brussels in 2023, The International Art Market Studies Association (TIAMSA) devoted its annual conference to the theme. Titled "Towards a Sustainable Art Market?", the conference gathered academics and researchers to debate sustainability in relation to the art market, i.e., to museums, curating practices, technology and media, and art galleries, among which this paper was first presented [8].

In spite of punctual measures, systematic change is needed. Participating in or being affiliated with organizations is an indicator of a commitment towards sustainability in the art market. Launched in 2020, the Gallery Climate Coalition (GCC) is an international community of art organizations that gathers professionals from different sectors, in addition to gallerists, such as artists and those from nonprofit organizations [9]. It aims to disseminate the best practices, reduce the carbon footprint, and create a strong commitment of zero waste until 2030, in line with the Sustainable Development Goals of the United Nations 2030 Agenda. This commitment aims at a prompt reply by the sector to the climate crisis. Despite being recent, this platform has raised global interest, and that is why we will use it as a standpoint reference in this study. Frieze and Art Basel are both members of the coalition from the art sector business. Concerning galleries, first steps have been taken from the Iberian Peninsula, with adherence by six Spanish galleries (Galería Baro, Badr El Jundi, Galeria Alvaro Alcazar, Galeria Senda, Galeria ATC, and Galeria Lucía Mendoza) and one from Portugal (3 + 1 Arte Contemporânea), as well as four foundations, museums, and non-profit institutions from Spain (Foundation Ideasbio, Fundación TBA21, Guggenheim Museum Bilbao, and JOYA: AiR) and one from Portugal (Insights of an Eco Artist), three Spanish companies related to the art market (Stick no Bills, TAC7 and UreCulture), and a varied number of artists and other professionals [9].

Taking into account the growing adherence to the Gallery Climate Coalition, a progressive awareness of the problem is evident in different agents of the art market, particularly the gallery owners and the art fairs organizers. However, artists have been the pioneers in this reflection on sustainability, making artistic proposals accordingly, before this issue was deemed urgent. A large number of artists have included a high commitment to the climate emergency in their works and in their creative processes for a number of decades, such as the "ecological artists" involved with ecological and environmental art [10,11]. From Georgia O'Keefe to Ana Mendieta, from Hokusai to Olafur Eliasson, from Joseph Beuys to Alberto Carneiro, this concern is well noted. The work of numerous curators in their exhibitions also shows their interest in tackling this problem from the perspective of curatorial criticism.

Within the remaining agents from the art market, collectors appears to be the group that seems to be the most reluctant in reacting in favor of a change in attitude. Considering that collectors are major actors in the global art market due to their purchasing of works of art and caring for the maintenance of their collections, travelling worldwide to attend art events, and fostering the art business, it is our belief that these actors also need to be studied as their change in behavior may act as a further step towards the achievement of the sustainable agenda UN 2030. The art ecosystem encourages trips to participate in events, and rarely uses or proposes the use of recycled materials to avoid waste. The economic model of the art markets in which collectors play lacks sustainability. From this, questions arise: How can we promote a sustainable collecting attitude? Can we identify alternative models in line with the climate change goals?



In 2022, a survey of global collecting was launched, where the report, while examining trade patterns, asked high net worth collectors about the environmental impact of collecting. The conclusions underline a growing concern and a higher awareness regarding the need for the adoption of sustainable options for the acquisition and management of works of art [12]. Nevertheless, collectors plan to maintain a high travel rate to see and collect art, as they prefer to view what they want to buy in-presence. Nevertheless, they are "willing to pay a premium for sustainably" in order to "reduce the environmental impact of their purchases" [12]. According to the above-mentioned report, collectors are aware of the relevance of sustainability issues, but instead of a compromise with a change in behavior, they are willing to pay an extra amount to reduce the impact of their purchases, using payment as an argument to replace a sustainable collecting practice.

In this exploratory article, our main aim is to analyze living contemporary art collectors from Portugal and Spain in terms of their contribution to the achievement of sustainable development goals in comparison with other players in the art market ecosystem. Using primary data, an inquiry, reports, field research, informal conducted interviews, and published sources, we studied their commitment to a sustainable environment—such as participation in local art fairs or, preferably, moving to online platformsas well as their interest in artists' discussions and agents' activities choosing a green attitude. With awareness of the challenge that preliminary research presents, the purpose of this research is to bring this issue to the attention of a wider audience of scholars and academics, as much as to disseminate knowledge and influence concerning the need for a paradigm shift toward purchasing behavior in the art market and to metaphorically stimulate a slow collecting practice.

## 2. Materials and Methods

In terms of the materials and methods we used to obtain the results of this research, we first analyzed the available literature on the subject. As this is exploratory research, the literature about collectors' awareness of sustainability issues is still scarce, and is absent when referring to the Iberian Peninsula collectors. This case-study is the first approach in terms of sustainable collecting practices in this geographic region. We studied the sustainability concept to frame the subject, taking into account both academic and non-profit organization perspectives (Kagan [10], UN, Bourdieu [13]), and the collecting theoretical definition (Pomian, Alsop, Pearce, Baudrillard, Elsner & Cardinal) [14–19]. We also analyzed the slow collecting concept, a derivation of the slow movement, which presents a need for slowing down against frenetic productivity and the social pressure of speed [20]. We consulted several published reports, both national and international (McAndrews [2], Bicycle [3], the Report of the World Commission on Environmental and Development, NEMO [6]), and as many unpublished reports that the authors had access to (For the research, we have used two unpublished reports from Duarte's authorship [21,22], namely: "A performance do mercado da arte em Portugal. O setor do antiquariato e das galerias de arte 2021" (The performance of the art market in Portugal. The antiques and art galleries sector 2021); and "O Impacto da COVID-19 no setor do Antiquariato e das Artes em Portugal" (The Impact of COVID-19 on the Antiques and Arts sector in Portugal). They were both presented in a public session, promoted by LAAF-Lisbon Art and Antiques Fair (Lisbon, Cordoaria, 5 May 2022; and Lisbon, Cordoaria, 17 September 2021). We have also taken into account the specialized press (The Art Newspaper [7]). The secondary sources were discussed between the authors and confronted with primary data gathered through field research (survey, semi-structured interviews and informal conversations with art players, collectors, gallerists, and artists on the topic). The primary data collected in the field by the authors were complemented by information from secondary sources, analyzed with the support of theoretical references.

The data examined in Section 4 were taken from an anonymous survey sent to Portuguese and Spanish collectors through a Google Form, and the questionnaire was sent by email between 22 May and 4 July 2023. In terms of the sample gathered, the social category

of a collector is defined as a living and active art buyer, recognized as such by themselves, by peers, and by agents in the art galleries from the milieu where those collectors operate (Becker) [23]. The collectors were identified using sources such as the galleries' contacts, former projects in which the authors were involved (Duarte) [24], their participation in the only association of contemporary art collectors in the Iberian Peninsula, and the authors' knowledge of the field. Those contacts were validated by telephone (26–29 June, 3 July). The questionnaire was organized in different sections to further understand how collectors have been approaching sustainability issues. The questions generally required closed answers, and the sections are as follow: (a) socio-demographic data, with inquiries about country of residency, age, gender, and period of time collecting; (b) commitment with the climate changes, with questions focused on measures adopted to reduce climate impact as travels, online purchases or recycling materials derived from purchases, perspective on the art market behavior regarding the subject, acquisitions from artists engaged with climate changes, knowledge on the curators' work on the topic. the perception of how the art market looks at other forms of sustainability, such as financial support for artists and gallerists, application of Good Practices, legal sustainability such as the regulation of online trade or NFTs. The analysis was conducted using aggregate data.

The sample comprised a group of 39 active collectors, approached differently in both countries: through direct contact with all collectors in the case of Portugal (a total of 31 collectors were contacted by telephone, although 20 responded to the survey), given Dr. Duarte's familiarity with the Portuguese 'art world'; in the case of Spain, Dr. Pérez-Ibáñez had the support of the collectors' association 9915 *Asociación de Coleccionistas Privados de Arte Contemporáneo*, the only association of contemporary art collectors in the Iberian Peninsula, with an estimate of 76 members (of which 19 responded to the survey).

For this paper, we combine theoretical references with quantitative data.

### 3. Sustainability and Collecting: Framing Polysemic Concepts

Sustainability is a concept that has been discussed in academia and in society, through non-profit organizations, for more than thirty years. 'Sustainability' and 'sustainable development' have both been systematized in conceptual closeness; the first reflecting the thought that development is synonymous with growth in a balanced evolution, and the later expression is based on a tryptic direction, as "social justice, ecological integrity and economic well-being" [10]. The author, researcher Sacha Kagan argues for a cultural dimension of sustainability, and using Moacir Gadotti's words, argues for a "dream of living well; [...] a dynamic balance with others and the environment, it is the harmony among differences" [10]. From a similar perspective, the United Nations (1987) defined sustainability as development that meets "the needs of the present without compromising the ability of future generations to meet their own needs" [25]. The definition remains current, although climate change has made the issue urgent. People's actions should be oriented, conciliating their needs without endangering the needs of successors, and without being a threat to sustainable development. A balance is required between economic, social, and environmental factors that guarantees that our actions are sustainable in the long-term to ensure a more equitable and prosperous future for all. We may underline two major ideas on the sustainability concept: the first is the satisfaction of the "*needs*" of societies, and the other is the idea of a "*limit*", meaning that the needs of societies must take into account the needs of the future [26]—an accomplishment that requires an articulated policy.

With these concepts in mind, what is sustainability when applied to the collecting process? How can we conciliate the *needs* of the present with its *limits*, or the needs of the future from the collectors' practices perspective? The art market is often measured in terms of sales and financial transactions, in which collectors play a key role. The more sales, the greater the success for the actors involved. On the other hand, taking into account Pierre Bourdieu's lessons—in which art has intrinsic value beyond its economic worth as it carries symbolic, cultural, and historical contents, along with social significance [13]—the art market functions beyond financial asset, contributing to the promotion and preservation

of artistic and cultural heritage. More than art trade, the market configures a platform for artists, gallerists, collectors, and intermediaries to engage with art and culture. Thus, the *need* for a sustainable collecting practice is more than an economic issue.

As the balance between the *needs* of the art market and the *need* for a more sustainable collecting practice is a key question, our research will delve into how this issue is being considered in two countries: Spain and Portugal. As a main market driver, to collect (art) is a social activity of assembling objects of every kind—even intangible, when related with projects or ideas—that are temporarily or permanently removed from the economic circuit, protected, and possibly exhibited for public view [14]. To assemble those 'things' implies to accumulate in a systematic way, where normally the criteria is defined. Collecting may be different from acquiring in terms of its intentions, in terms of the creation of a narrative among those objects. Nevertheless, in the frame of a behavioral and cultural system, it lacks a sufficiently precise definition in the literature to be universally useful [16], driving the need for the proper identification of the collection by the collector himself or herself [17]. Motivations are also decisive to the collecting process, varying between aesthetics, financial, and social engagement. From a psychological perspective, the relevance of collecting counteracts the fleetingness of life through the perenniality of works of art [18,19].

The idea of assembling that characterizes the collecting process may contradict the sustainability concept when it argues for a behavior of reusing, recycling, a slow collecting process, or a responsible collecting method. The art market stimulates consumption and accumulation; therefore, paradoxically, discussing sustainability and the slowing down of the collecting behavior challenges its intrinsic rules.

Recovering the idea of the *limits* embodied in the previous definition of sustainability from the UN, by which collectors should take into account the needs of the future, collectors seem to adopt a superficial perspective in their practices, as we will see below. Some of them are choosing green, indeed, by adopting actions towards a low footprint carbon, showing concern about it and are willing to help to resolve it. But systematic actions are needed. Sustainability encompasses different dimensions, implies long-term commitment, a change of conscience, and, therefore, a paradigm shift in attitude with the power of exerting influence over others. Before returning to this perspective, we will show what institutions, galleries, and professionals are doing in terms of their sustainable practices.

### 3.1. The Needs and the Limits: Sustainability Measures from Institutions, Galleries, Professionals

The commitment to climate sustainability that the Spanish and Portuguese art system are demonstrating is very recent and not yet widespread, with few museums, galleries, and public and private institutions involved in this problem, let alone collectors, as we will see in the next heading. The active measures to stop climate change are concentrated, above all, in the activity of the Gallery Climate Coalition. As the organization states in its web site [9]:

> GCC's primary goal is to facilitate a reduction of the sector's $CO_2e$ emissions by a minimum of 50% by 2030, as well as promoting zero waste. We develop and share best practice, provide leadership on sector specific environmental issues, and work to leverage the collective power of our membership to achieve systemic changes.

Among their members, some stand out, especially those involved in the adoption of active measures. Below, we will emphasize some initiatives that have recently been developed by agents from the art market, such as museums and curators, galleries and art fairs, and alternative business models, to provide perspective and compare it with the collectors' commitment.

#### 3.1.1. Museums and Curators

The TBA21 Foundation—Thyssen-Bornemisza Art Contemporary, through a curated program of exhibitions in different venues and the research activity of the TBA21 Academy, seeks to reinvent the culture of exploration in the Anthropocene of the 21st century, while developing knowledge creation, new modes of collaboration, and the co-production of solutions for the pressing environmental challenges of today. With exhibitions such as

"*Futuros abundantes*" or "*Remedios. Donde podría crecer una nueva tierra*" [27], and with the close connection with committed artists like Joan Jonas, Olafur Eliasson, Shirin Neshat, Dominique Gonzalez-Foerster, and Tomás Saraceno, and curators like Stefanie Hessler, Chus Martínez and Daniela Zyman, TBA21 raises an approach to the problems arising from the climate emergency and the overexploitation of resources starting from creativity, technology, ancestral traditions, and respect for our origins.

As for the Guggenheim Museum in Bilbao, coinciding with the celebration of its 25th anniversary, it presented an ambitious sustainability plan, a pioneering approach in the museum sector, aimed at minimizing the environmental impact of its activity through the implementation of energy-sustainable solutions and non-polluting processes, and promoting activity oriented towards eco-efficiency. These measures are intended not only to guarantee compliance with environmental regulations from all legislative areas and the obligations derived from permits, licenses, etc., but also to expand, as far as possible, its commitment. In 2004, the museum obtained certification corresponding to the implementation of an environmental management system in accordance with ISO 14001 [28], which has allowed it to implement a structure to control the environmental impact of the museum's activity and services and maintain effective management to continuously improve its performance. Among the measures adopted, the museum cites the following:

- Energy optimization and reduction in electricity and gas consumption;
- Selective waste collection and recycling;
- The authorization of a specific warehouse for the management of hazardous waste;
- The reduction in sound impact in the environment;
- The re-direction of the drainage network of discharges (except rainwater);
- Carrying out annual external and internal environmental audits;
- Training and awareness communications for the workforce.

In addition to its direct emissions, the museum has calculated its carbon footprint for a substantial part of its indirect emissions during 2019, and the total figure amounts to 4313 tons [29]. The calculation of the museum's indirect emissions, known as 'Scope 3', is key to identifying opportunities for energy efficiency and savings in its daily activity. It is the first museum, at the international level, to take this step. The museum has calculated and certified its carbon footprint related to the transport of works of art and the movements of staff, which corresponds to a third of the total emissions. As a result of this verification, the museum has consolidated some initiatives and is going to implement new actions to reduce the consumption of energy and raw materials involved in the organization of its exhibitions.

Guggenheim Museum Bilbao will also participate in the Symposium on Environmental Sustainability, organized by the World Art Foundations WAF, in Bilbao, in October 2023. As the website reads, this symposium offers "a practical approach for decision makers of Art Foundations to develop and implement environmental strategies within their organizations" [30] with a clear urgency around the climate crises.

Outside the ranks of GCC, we find private art institutions and foundations, other professionals, and companies related to the art market that show the same commitment to sustainability, albeit in other ways. The Serralves Foundation in Oporto has been developing a program of good practices in terms of climate concerns, articulating a double mission of disseminating contemporary values with environmental issues. Within this frame, since 2011, it signed a co-operation protocol with the Portuguese Environment Agency for the implementation of a Community Eco-Management and Audit System (the EMAS aims to promote the efficient management of the environmental system) [31]. The same concern is noted by the Calouste Gulbenkian Foundation, in Lisbon, which developed a program, beginning in 2018, titled *Sustainable Gulbenkian*, and supports initiates and programs to stop climate changes [32]. It launched (June 2023) the "life cycle evaluation" of "Europa Oxalá" (Europe Hopefully), a temporary exhibition with contemporary artists chosen due to their familiar origins in the ex-African colonial countries to discuss post-

colonial Europe, in 2022. Its significance lies in the fact that it is intended to be a model for the evaluation of other exhibitions in terms of reducing the carbon footprint [33]. In the conclusions, it pointed to the displacement of people (visitors, artists, curators, journalists) as the main factor that contributed to the carbon footprint, in the order of 61%, although the Gulbenkian Museum is paradoxically served with very good access to public transport.

Museums are players that are particularly positioned to have a key role in raising public awareness about sustainability as they are meant to serve societies with an inclusive profile, fostering diversity and sustainability [34]. The former aim was recently added to its definition during ICOM's General Conference in Prague in 2022, making the museum a leading institution in offering education in favor of a sustainable commitment.

The work of curators such as Blanca de la Torre, director of the 15th Cuenca Biennial in Ecuador, entitled "Biocene Biennial. Changing green for blue", is closely linked to the need for sustainable and ethical processes at all levels. Similarly, artists like Eugenio Ampudia, with his multidisciplinary work "Concert for the Biocene", invites us to reflect on our relationship with nature through music.

### 3.1.2. Galleries and Art Fairs

Galleries, as private commercial structures where works are displayed [35], also seem to want to show their concern towards climate sustainability not only through the active practices and measures proposed by GCC, but also by working with artists who are committed to criticizing over-exploitation policies and the need to raise social awareness about this problem. We can highlight Lucía Mendoza gallery in Madrid, with the work of Elena Lavellés, Miguel Sbastida, Luna Bengoechea, Agustín Ibarrola, Lucía Loren, or the Portuguese 3 + 1 Arte Contemporânea in Lisbon, with artists such as Adriana Proganó, Alberto Carneiro, and Evy Jokhova.

Concerning the art fairs held in the Iberian Peninsula, the main commercial venue in the primary art market [36], the ARCOmadrid art fair, and its subsidiary, ARCOlisboa, have also developed environmental sustainability policies. Specific actions are being carried out, proactively, that affect the reduction in any type of consumption: paper, wood, plastic, metal, or textile (they do not use carpet by default, entrance tickets have been physically minimized, and those that remain are no longer PVC). Online registrations in the Collectors and GUEST Programs have also been promoted, and all of these programs are digital. On the other hand, the modular booths are easily removable and reusable, and their composition allows for their recycling. In addition, their weight has been minimized to the essential minimum, without compromising safety and avoiding the use of dangerous chemical products. The lighting of all the modular booths is produced by led fixtures. In terms of food consumption, for example, ARCO communicates daily visitor forecasts so that catering companies can prepare the appropriate supply. ARCO promotes the use of collective transport to transfer all the guests from their accommodation to the art fair (round trip,) between approximately 600–800 people every time. There are also electric car sponsorships.

ARCO, as well as Estampa, the art fair that takes place in Madrid every autumn, is part of the public institution IFEMA Madrid and aligns with the organization's sustainable strategic management plan [37]. Therefore, it:

- Prepares and implements, at least annually, a CSR Plan, with the lines of action and the actions to be carried out in each of them and their relationship with the SDGs;
- Sustainability is part of IFEMA's culture as one of its values;
- Sustainability is part of IFEMA's Strategic Plan.

Regarding greenhouse gas emissions, IFEMA annually calculates its scope 1 and 2 carbon footprint, requesting the corresponding seal and updating the register in the MITECO (Ministry of Ecological Transition). It has reduced its footprint by 90% in the last 6 years.

IFEMA has implemented the best techniques available regarding the consumption of electricity, natural gas, water consumption, and paper consumption:

- Certified 100% renewable origin electricity;
- Installation of geothermal energy in the Puerta Sur building;
- Low-consumption lighting in halls and modular stands;
- Efficient irrigation systems and irrigation with reclaimed water;
- Electronic refrigeration, dry urinals, double flush toilets;
- Paper from sustainable forests (FSC or PEFC certification).

Regarding waste regeneration:

- Provides clear information on waste management regulations (e.g., signs, public address system, written regulations, etc.);
- Offers a service to exhibitors that includes the segregation of the generated waste and its management until its recycling;
- Makes rubbish bins near its exhibition area available for the event to deposit waste in a segregated manner;
- Has reduced its waste generation ratio by 59% in the last 5 years;
- Has recycled 83% of the waste generated in 2022.

Finally, as we have mentioned about ARCO, in terms of transportation:

- IFEMA has public transport close to its facilities;
- Has parking for electric cars with 100% renewable electricity supply and parking for bicycles;
- Has internal transport (bus or similar) that uses alternative energies (electric, gas, biodiesel, or hybrid vehicles, etc.);
- Promotes collective transport and public transport from hotels to the fair for visitors and exhibitors.

It is noted that both galleries and art fairs seem aware of the current sustainable challenges. Both galleries and art fairs are developing initiatives accordingly, mainly concentrated on the reuse and recycling materials. However, there is still a lack of studies on the results of these actions, evaluating and assessing their effective impact.

### 3.1.3. Alternative Business Models

We may also refer to alternative business models, which function in a complementary way to the art market's mainstream structures. In this case, we will return to the polysemy of the concept of sustainability, expanding its meaning towards the search for a social, financial, and ethical balance between the different agents that make up the art system. In Portugal, an online platform, The Concerned Collector [38], and a new art fair, the Mercado P'la Arte [39], both opened during the pandemic health crisis context, and despite not assuming in their mission guidelines clearly aimed at climate change, their inclusive mode of action in the art system helps to put into perspective the scope of this subject. The first is an online auction with artworks from living artists, whose works are chosen in artists' ateliers from those unsold in the art galleries. This business method does not create concurrence with galleries, nor does it interfere with artists' quotation. In addition, 100% of the auction benefits go to the artists. The management of the platform takes a percentage, but only from the buyer. The second is an art fair, which has the support of a non-profit organization, P'Arte, with the novelty of having no intermediaries in the acquisition process, e.g., artists are in-presence monthly in a garage, in Lisbon, exhibiting and selling their works. These initiatives are meant to dynamize the art market, fostering a non-speculative business through the practice of low prices, thus contributing to financial sustainability of the artists. In addition, this trade modality provides access to art to people that were not necessarily familiar with the artistic milieu.

Aligned with these measures, the Spanish art market became aware, after the 2008 economic crisis, of the need to commit to financial and social sustainability, and to an art market suited to the new business models that were proliferating [40]. The need to adapt to a less local, more global collecting that arose during that period and the decrease or disappearance of public aid or institutional collecting that, until then, had contributed to the

maintenance of the commercial activity of many galleries, impelled new gallery owners and the existing ones to look for ways to maintain their activity and to recover the viability of the galleries. The new circumstances of the global art market, determined by the development of new technologies and by the birth of a new type of collector, made it necessary to modify the strategies that had been used to date. The new galleries that proliferated in the Spanish market responded, therefore, to a different style. Delocalized, ubiquitous, multidisciplinary models that accommodated parallel exhibition projects focused on complementary market niches, which allowed gallery management to be sustainable. The galleries that emerged then, with a strong global component, such as Sabrina Amrani or Travesía Cuatro, or with a pop-up model of great international artists, such as Bernal Espacio, or diversified in terms of their own business model, such as Swinton and Grant or Delimbo, have continued to demonstrate sustainable models following the pandemic.

In the case of artists, the generation of independent, alternative, and self-managed spaces enabled the survival of numerous artists who lacked representation by galleries—spaces that we can define as "third place", the concept coined by the sociologist Ray Oldenburg (1989), halfway between the "standard" workspace and the place of exchange where artists combine similar backgrounds and objectives with their own patterns and creative, stimulating, and enriching processes [41]. These spaces were true laboratories for research, development, production, and artistic residences, combined with the administration and marketing of works of art, and meeting places with the public and with other agents of the art system (galleries, collectors, institutions, managers, critics, curators, other artists). Good examples of this model are Mala Fama and Nave Oporto in Madrid, and Escalera de Incendios and Chiquita Room in Barcelona.

This shows that private art institutions, alternative companies, professionals, and the most important art fairs in the Spanish and Portuguese capital are taking active measures in favor of sustainability, both environmental and social/financial, and are also stimulating the participation of galleries to compromise with sustainability values. What about collectors?

## 4. Results: Collectors and the Lack of Awareness for a Sustainable Collecting

In the introduction of this study, we raised the possibility of collectors being the social group that reluctantly reacted in favor of a change in attitude more suitable for the sustainability concerns. Indeed, our prospecting work approaches the measurements from the perspective of collecting, especially from private collecting (In a personal interview carried out by Pérez-Ibáñez, the president of 9915 association expressed his unfamiliarity about the measures that were already being taken by institutions, art fairs and galleries, and his doubts about the need to take part from private collecting, which has been ratified in the results of our survey). The environmental commitment of the institutions, as well as of the corporate and public collections, is closely linked to the legal imperatives by which the Ministries of the Environment are imposed to follow certain guidelines. For this reason, it was especially interesting for the authors to know whether the private initiative, or private collectors, shared this commitment and took measures accordingly. We have also approached other modes of social and financial sustainability, including legal, to which the consulted collectors contributed their opinion.

In order to obtain the opinion of a number of Spanish and Portuguese private collectors, a brief sample was taken, and a survey was launched that had to be completed online, as we explained in the Section 2 of this paper. The survey, which was active for two weeks, was answered by a total of 39 people from both countries, very evenly distributed, as Figure 1 demonstrates.

The gender and age of the surveyed collectors also showed a usual trend, according to the reports on collecting in both countries: a majority of men compared to women and a majority of ages over 50 years, having no respondents younger than 30 (Figures 2 and 3) [21,22]. We may mention that all the collectors participating from Portugal were men.

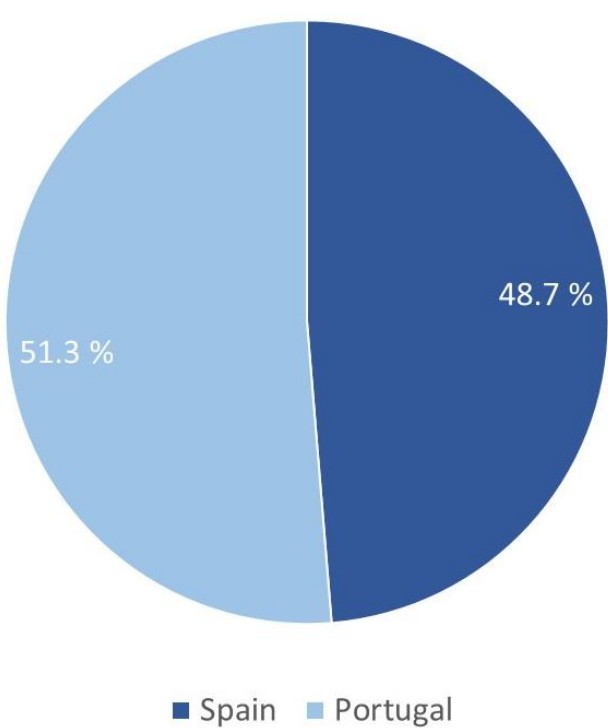

**Figure 1.** Collectors' country of residency. Data source: Survey conducted by authors.

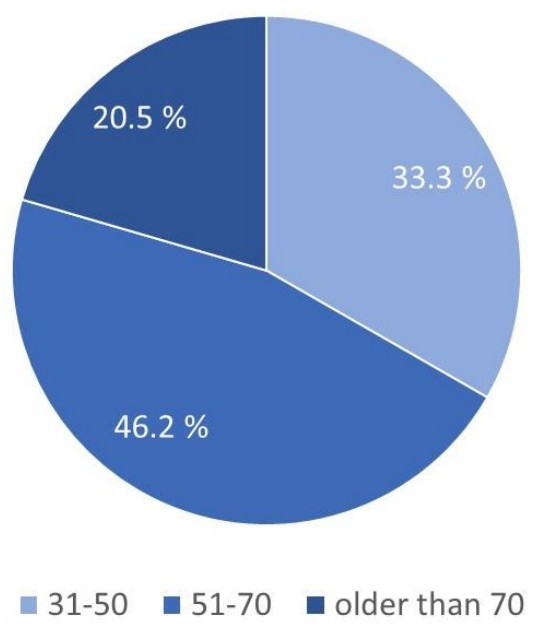

**Figure 2.** Collectors' age. Data source: Survey conducted by authors.

The number of years dedicated to collecting was also within expectations, giving priority to collectors dedicated to this activity for over 20 years (Figure 4.)

Regarding the collectors' commitment to the climate change problem, only 38.5% responded positively, and only 35.9% declared that they took measures to reduce the climate impact of their activity as a collector (reduce air travel, switch from on-site purchases to online purchases, increase recycling and reuse of materials derived from purchases.)

The collectors in our study were also asked if they considered that measures were being taken in this line by the art market, galleries, and art fairs. To this question, only 15.4% answered affirmatively. The remaining 84.6% did not report any commitment from the art market for this problem.

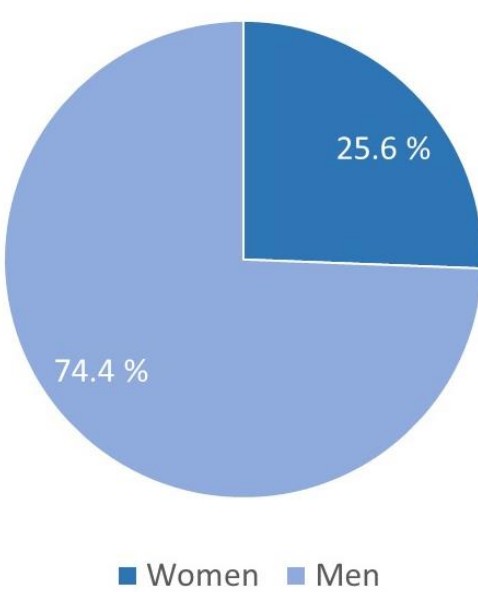

**Figure 3.** Collectors' gender. Data source: Survey conducted by authors.

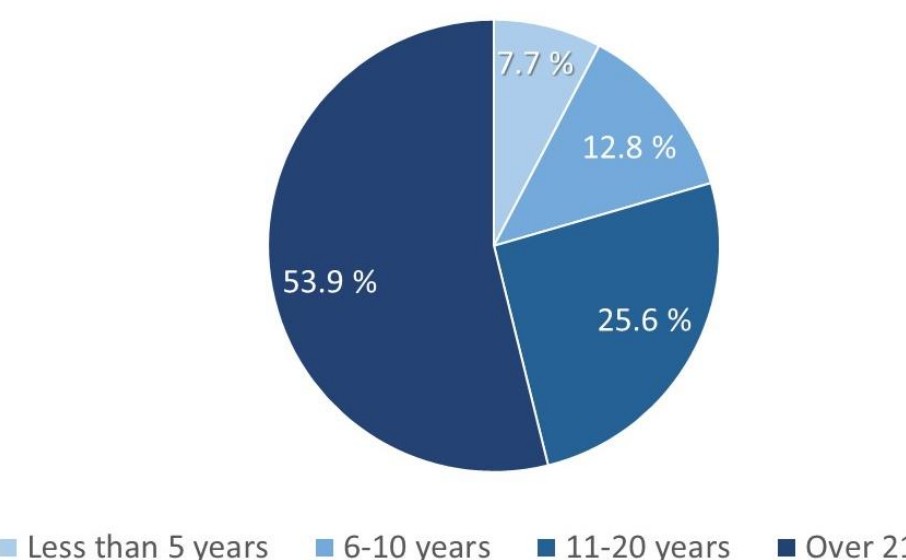

**Figure 4.** Period of time collecting. Data source: Survey conducted by authors.

They were also asked if they considered that the art market was sensitive to other areas, such as financial sustainability (support for artists and galleries) or the application of Good Practices or legal sustainability (strengthening of regulation on digital art, of NFTs market, international market protection). In the first two cases, the collectors were optimistic; they were less so in the third (Figure 5.)

Regarding the last reflections from the surveyed collectors, it is mentioned that the commitment to sustainability is transversal; that is, the world of the arts is not left out of this topic. It was also commented that it has been used as an advertising "flag", but with little scientific basis, and above all, with little result, emulating a sort of greenwashing effect, and that awareness for climate change has not yet permeated the art market; that there are committed artists and curators, but that it does not seem to be a very visible issue yet. One collector stated that the lack of transparency in the art market affects the confidence of collectors and the sustainability of the market, as well as the support for artists who can oppose political correctness, distinguishing between art and social militancy.

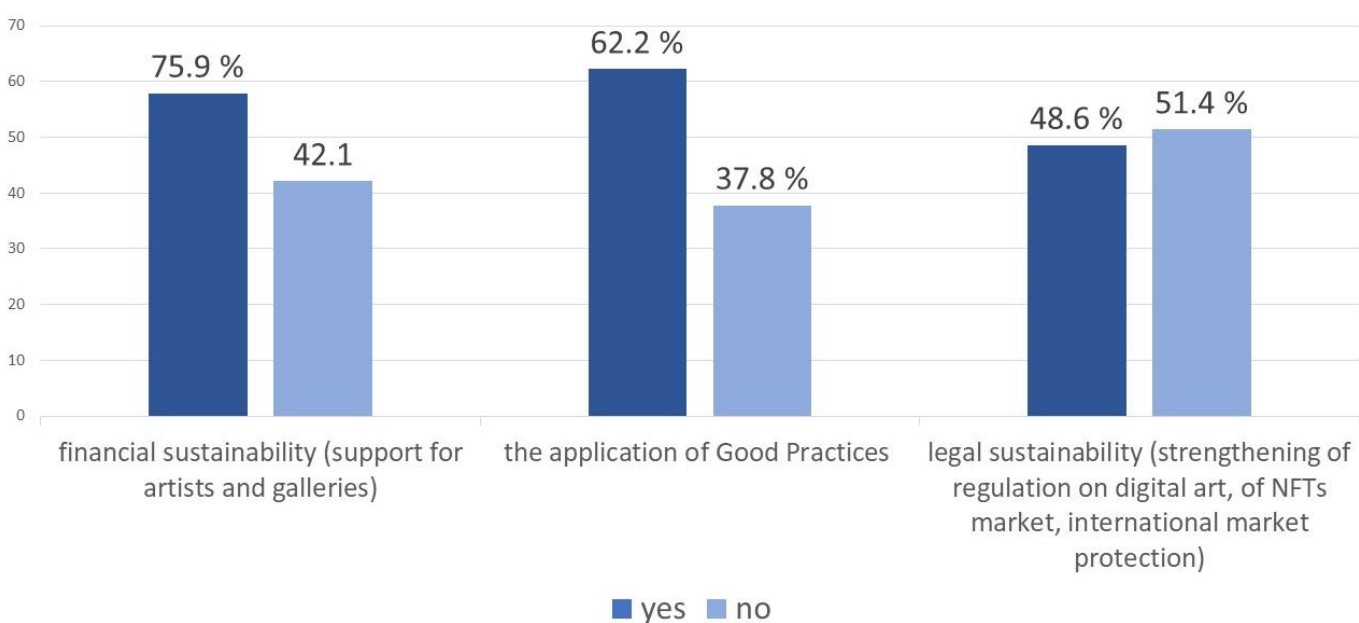

**Figure 5.** Collectors' commitment with different areas of sustainability: financial sustainability (support for artists and galleries), the application of Good Practices or legal sustainability (strengthening of regulation on digital art, of NFTs market, international market protection). Data source: Survey conducted by authors.

In general, the collectors we approached stated that in the Spanish and Portuguese markets, there is more awareness for regulation, economic sustainability, and viability than for climate awareness, and the lack of interest in this issue among collectors is evident.

## 5. Discussion: Slow Collecting and The Need for a Paradigm Shift

The results of the survey were not optimistic in terms of active measures towards sustainability and to stop climate change. As we previously noted, the majority of the surveyed collectors (61.5%) did not claim a commitment to climate change issues in their collecting practices.

Nevertheless, it is worth paying attention to some of the comments they made in the frame of this inquiry because it opens the sustainability concept to a broader perspective, reinforcing the previous idea of a polysemic concept. When asked to share their thoughts on the topic and to point out the kind of measures they have taken, in addition to mentioning the recycling of materials, online acquisitions (although the increase in online acquisitions is not clearly an aim, it helps to diminish the effects of climate crisis), reducing in-presence art fairs attendance, and the optimization of the conditions of collections (temperature and humidity), the collectors also mentioned local acquisitions, the relevance of transparency for their own confidence and for market sustainability, the support of committed artists, the need for more adequate regulation, and the financial and the legal balance. This means that sustainability is more than a sum of initiatives related to recycling, carbon emissions, reducing plane travel, and the optimization of resources. It is a behavioral change, a paradigm shift, an experimentation of alternative models of the art market functioning and collecting dynamics.

The commitment to sustainability is observed in alternative and unexpected paths, new business models, innovative and collaborative strategies [42], and in small-scale trade structures. The sample of collectors from the Iberian Peninsula under analysis tend to acquire human-scale business, such as through local art fairs and galleries, preferably from local artists; some of them committed to sustainable materials or techniques, as well as underrepresented communities, helping them to preserve cultural diversity, a tendency that contradicts—or complements—the global art market. Whether or not these actions



are motivated by sustainable climate change reasons, taking into account the polysemic concept, this tendency is definitely relevant for financial sustainability.

Approaching an answer to the question raised in Section 3, *what is sustainability when applied to the collecting process*, we think it could be the idea of slow collecting—a metaphor for a responsible collecting process, for a change in behavior that claims an articulation within the art market structures and within public policies. The slow collecting category is a derivation we appropriate from the slow movement that occurred in the 1980s, demanding reflective behavior, a deceleration of the frenetic productivity, and valuing quality instead of quantity in a balance between the local demands and the global aspiration [20]. Slow collecting engages deeper with artistic communities, invests time to understand the local context, develops a continuous practice of support, and shows concern for the environment and the art ecosystem. Engagement with sustainability takes longer, suggests a need for a network, and includes the adoption of political policies and ethical approaches that include transparency in the art market and the acceptance of the Good Practices towards an inclusive art world.

Collectors will probably not reduce their travels, and nor will they change their acquisitional behavior or their taste on a sustainable premise. But some measures taken already during the pandemic lockdown, such as the turn to the online art market and to local galleries and artists, together with the recent awareness of the role collectors may take in considering climate change as one of the issues our society must face and solve, are resulting in a new way of dealing with collecting while being concerned with social and environmental problems. Regarding the fight in favor of social and financial sustainability in the art market, the economic crises have been key to detecting the need to establish measures that favor a more ethical and viable market for all of the agents involved. We trust that, in the future, the measures that have proven to be reliable and positive will strengthen sustainability in all of its aspects.

**Author Contributions:** Conceptualization, A.D. and M.P.-I.; methodology, A.D. and M.P.-I.; formal analysis, A.D. and M.P.-I.; investigation, A.D. and M.P.-I.; resources, A.D. and M.P.-I.; data curation, A.D. and M.P.-I.; writing—original draft preparation, A.D. and M.P.-I.; writing—review and editing, A.D. and M.P.-I.; visualization, A.D. and M.P.-I. All authors have read and agreed to the published version of the manuscript.

**Funding:** This research received no external funding.

**Institutional Review Board Statement:** This study did not require ethical approval.

**Informed Consent Statement:** Informed consent was obtained from all subjects involved in the study.

**Data Availability Statement:** Google Forms platform: https://docs.google.com/forms/d/1y5n_OQdnLrVZ3ke5GNJdvXBu97m0Pfx4ROOyYWRPLjo/edit#responses (accessed on 4 October 2023).

**Conflicts of Interest:** The authors declare no conflict of interest.

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
