# Peer review of "Slow Collecting: Sustainability and the Need for a Paradigm Shift by Iberian Collectors"

_sustainability, doi:10.3390/su152115401_

Round 1
Reviewer 1 Report
Comments and Suggestions for Authors
The work is very interesting and original, and an important contribution to the reflection and awakening about sustainability in the art collectors' circle, through an analysis in the Iberian Peninsula context. However, I would suggest a few minor considerations to make the work's contributions more explicit and to better appreciate its richness.
In section 2, on materials and methods, it is stated that the sample of collectors was 39, as mentioned in section 4. However, in section 2 it is suggested that the sample in Portugal was 31 collectors, implying that the number of collectors in Spain would be 8. This would show a discrepancy between the samples. In the case of the Spanish sample, it appears to be an even smaller sample when compared to the number of collectors in the association being analysed (73). However, in this case, only 8 were included in the sample; and since Spain is a much larger country than Portugal in terms of area and population, it would be expected that the sample would therefore be larger. Despite this assumption, section 4 states that the samples are "very evenly distributed", as shown in the first graph. It would be very important to clarify this issue so that it is clear exactly how many collectors there are in each country in the sample.
Some small suggestions are made in the comments on the attached document.

Author Response
Thank you so much for your comments. We are pleased that you found this paper an important contribution to the reflection and awakening about sustainability in the art collector’s circle.
Concerning the issue you raised, on clarifying the sample of collectors who participated in our study, we have done it, on lines 164 and 168-169, and we hope it is clear now.
Plus, we added information on the two unpublished reports, as asked, and changed for perenniality.

Reviewer 2 Report
Comments and Suggestions for Authors
The present study was aimed to analyse living contemporary art collectors from Portugal and Spain, in terms of their contribution to the achieving of sustainable development goals, in comparison with other players of the art market ecosystemat. The work is original. The scientific quality of article is quite high. The materials and methods used in this study are well described and presented. The obtained results are well presented and interpreted. In my opinion, the article can be published after some minor revisions to further improve its quality.
1. The quality of all figures should be improved.
2. The legends for all figures must be written in English.
3. The reference list should follow the journal style.
Author Response
Thank you so much for your comments. The amendments asked concerned images were made in the article – the legends of all figures are in English.
Concerning the reference list, we followed the journal recommendation, that is the references must be numbered in order of appearance in the text and listed individually at the end of the manuscript.

Reviewer 3 Report
Comments and Suggestions for Authors
It is a good article and some correction points are highlighted in the text of the article.

Author Response
Thank you so much for your comments that allow us to try to be clearer.
To respond to your queries, we will start by mentioning that in the abstract we consider that we have identified our problem, which is the lack of awareness of one of the major players in the art market system, the collector; then, we pointed the methods we used and the materials to carry on the research of our problem, the analytical method of primary and secondary sources; and finally, we have addressed our conclusion, that is the need for a paradigm shift and for a slow collecting attitude.
Concerning the research gap (section 2), we think we have identified it, putting into perspective the paradoxical situation of the art market system and the main players, particularly collectors that, for one hand, showed concern with sustainability challenges, but, on the other, they kept the will of keeping the same behavior. Then, we explain how we will proceed with research to achieve our main goals.
The results section (4) - collectors and the lack of awareness for sustainable collecting - intends to show the results of our primary data, that is the survey we launched to collectors from the Iberian Peninsula. We based the conclusions of our study on the primary data obtained at the survey, in section 5) discussion of the need for a paradigm shift. Through it, we respond to the main research goal, which is “to analyse living contemporary art collectors from Portugal and Spain, in terms of their contribution to the achieving of sustainable development goals, in comparison with other players of the art market ecosystem” (lines 111-113), mentioning that the “results of the survey were not optimistic in terms of active measures towards sustainability and to stop climate change” (line 496). But, at the same time, the information obtained, allowed us to interpret the concept of sustainability as a polysemic one, as we stated “This means that sustainability is more than a sum of initiatives related to recycling, the carbon emissions, reducing plane travels, and optimization of resources. It is a behavioral change, a paradigm shift, an experimentation of alternative models of the art market functioning and collecting dinamics”. (lines 511-515). This is why we concluded that the “idea of slow collecting, a metaphor for a responsible collecting process, for a change in behavior that claims an articulation within the art market structures and within public policies”.
Finally, the questions of the survey written in Spanish and in Portuguese – addressed to collectors in their native tongues – are repeated in the article in English, so that readers, who are not familiar with those languages, can follow the discussion and the arguments in that section.
